# Random Vibration Fatigue Analysis Using a Nonlinear Cumulative Damage Model

**Jesús M. Barraza-Contreras \***, **Manuel R. Piña-Monarrez**, **Alejandro Molina** and **Roberto C. Torres-Villaseñor**

Industrial and Manufacturing Department, Engineering and Technological Institute, Universidad Autónoma de Ciudad Juárez, Ciudad Juárez 32310, Chihuahua, Mexico; manuel.pina@uacj.mx (M.R.P.-M.); al187118@alumnos.uacj.mx (A.M.); al153286@alumnos.uacj.mx (R.C.T.-V.)
\* Correspondence: al187061@alumnos.uacj.mx; Tel.: +52-656-330-1229

**Featured Application: To apply the proposed model and its method, the inputs required are vibration power spectral density (PSD) and material characteristics, thus the bending stress provoked by vibration can be determined by using response acceleration. The proposed model incorporates the damage induced by the stress of the random vibration, then, the fatigue life is estimated. Thus, for further research, the model can be used to formulate fatigue analysis considering a material's strain or crack growth.**

**Abstract:** The paper's content allowed us to determine the fatigue life of a component that is being subjected to a random vibration environment. Its estimation is performed in the frequency domain with loading frequencies being closer to the system's natural frequency. From loads' amplitude and their interaction effect, we derive a nonlinear damage model to cumulate the generated fatigue damage. The exponent value of 0.4 from the Manson–Halford curve damage model was replaced by a vibration bending stress relation that considers the effect and interaction of loads. The analysis is performed from a progressive accelerated vibration spectrum to predict the fatigue life estimation. From this accelerated scenario, the accelerated coefficients and cumulated damage are both determined. The proposed nonlinear model is based on the following facts: (1) vibration and bending stress $\sigma_{vb}$ values are obtained from the response acceleration of power spectral density (PSD) applied and (2) the model can be applied to any mechanical component analysis where the corresponding acceleration responses $A_{res}$ and the dynamic load factor $\sigma_{dynamic}$ values are known. The steps to determine the expected fatigue damage accumulation $D$ by using the curve damage are given.

**Keywords:** fatigue damage; random vibration; resonant frequency; acceleration response; non-linear accumulative model

## 1. Introduction

Several systems and components may be subjected to vibrations during their operational life. Random vibration induces fatigue damage by dynamic loads and their amplitudes [1], which causes deflection in the component. The maximum experienced stresses are generated as a response of the natural frequencies of the component [2], mainly when products are operating close to those natural frequencies (resonance frequency). Thus, components must be designed to withstand the induced fatigue damage. Therefore, during product development, it is necessary to validate the component functionality through durability/validation tests. Furthermore, since predicting the fatigue damage is a complex process [3], nowadays, its accurate prediction is a fundamental engineering problem. For that purpose, some cumulative damage models have been proposed; some of these include the modified Steinberg vibration lifetime model [4] that considers that the effects of the vibration are accurate predictions. The synthesis of sine on random vibration based on fatigue damage spectrum [5] preserves not only the induced fatigue damage but

also the deterministic conditions of the environmental vibration. Additionally, there is the Dirlik's method [6], which is based on empirical closed form expression of the probability density function (PDF) of the rainflow amplitude, that, in the analysis, employs the stress amplitudes and coefficients functions of the power spectral density PSD of the stress. Various other frequency fatigue damage models have been developed and reviewed [7], however, at the phase of determining the damage accumulation, most of them use the linear damage rule.

The linear damage accumulation rule, also known as Miner's rule, is widely used in fatigue life prediction analysis [8]. It is given by

$$D = \sum \frac{n_i}{N_1} = 1 \qquad (1)$$

where $D$ is the fatigue damage, $n_i$ represents the number of the cycles of applied load at a given stress level $\sigma_i$ and $N_i$ are the number of cycles to failure at $\sigma_i$. However, this model does not consider changes in the effect of stress and does not consider the load sequence of application of stress in the component [9].

In practice, the fatigue damage induced in the mechanical components can be analyzed based on stress, strain and crack growth rate [10]. Moreover, according to [11,12], the effect of the stress is an important parameter that address the fatigue lifetime of the component. In this research, the analysis is performed based on the stress approach by using the random vibration PSD function of the stress response. Because the vibration stress is generated by the variable amplitudes of the cyclical loads, then it is not recommended to estimate the vibration fatigue lifespan by a linear analysis that is independent on the generated response stress level and without considering their interactions [13]. Consequently, since fatigue damage is a complex process that involves many factors [14], then a nonlinear damage analysis is recommendable [15,16]. For these reasons, in this paper, we present a modification of the damage curve model [17] that includes in its assessment of the damage accumulation the interaction effect load and the sequence load of the random vibration applied. Since the inputs for this nonlinear proposed model are the response acceleration values that provide the bending stresses by using a dynamic factor and by replacing the constant exponent value of 0.4 on the damage curve with a response stress relation, the model's efficiency is that it can be applied to any mechanical element to estimate and perform an analysis of the fatigue damage accumulation induced by a PSD of random vibration.

To show numerically how the proposed model works, in a practice analysis, the response stress generated from the loading random vibration is obtained by determining a dynamic load factor ($\sigma_{dynamic}$) and the corresponding response acceleration ($A_{res}$) factor. Then, based on the generated vibration bending stress ($\sigma_{vb}$), the total life cycle ($N_i$) is determined by using the Basquin's equation. The cycles of random vibration applied ($n_i$) are determined by using the rainflow method. Thus, with these three factors, vibration bending stress ($\sigma_{vib/bend}$), the total cycle ($N_i$) and the cycles of random vibration, applied ($n_i$), the modified model is performed. Since the proposed model depends only on the vibration bending stresses and its analysis to determine them, then, in the application, a single mechanical component (support) made of steel AISI 1025 subjected to random vibration (bending cycling) stress was used. The addressed vibration bending stresses were determined once the component was submitted to the random vibration profile per GR-326, then, by applying the proposed model, the fatigue damage until failure ($D = 1.00$) was determined after 28 loads of the GR-326 vibration profile.

The structure of the paper is as follows. Section 2 presents the theorical background of the random vibration and nonlinear damage accumulation. In Section 3, the proposed nonlinear model is formulated. Section 4 contains the experimental application (validation's procedure) and its result. Finally, the conclusions are provided in Section 5.

## 2. Theorical Background

One of the most common analyses to represent a dynamic response of the random vibration load input is by using the frequency domain, which implies the use of the PSD [7].

### 2.1. Random Vibration

In the frequency domain, the random vibration behavior can be well represented with a PSD function. Random vibration allows products to resonate all of the time [18], and by its cyclic movement, it causes cumulative fatigue damage, which results in vibration stress that reduces the strength of the material to support it. Thus, due to the generated strain, the repetitive vibration stress produces the failure of the component [19]. In the analysis, we must consider that the product being vibrated responds to the input vibration as a function of the product's resonant frequency [18], where the amplification factor of the response $Q$ is expressed as the transmissibility of the vibration amplitude at the resonance frequency, and it is given by,

$$Q = \frac{f_n}{\Delta f} \tag{2}$$

where $f_n$ is the natural frequency in Hz, and it is determined either by Equation (3) or Equation (4),

$$f_n = \frac{1}{2\pi} \sqrt{\frac{k}{m}} \tag{3}$$

$$f_n = \frac{W_n}{2\pi} \tag{4}$$

$m$ is the mass. $k$ is the material's stiffness and $W_n$ is the natural frequency in Rad/s. Here, we highlight that the $W_n$ value included in Equation (4) is determined according to the geometry, support and loads of the component under analysis. Once the component is submitted to the stress of random vibration PSD load, the generated fatigue damage is accumulated in a nonlinear way, as follows.

### 2.2. Nonlinear Fatigue Damage Accumulation

Regarding the nonlinear fatigue damage accumulation, Richart and Newmark [20] presented a damage curve model. Then, based on it, Marco and Starkey [21] developed the first nonlinear load dependent damage accumulation model as,

$$D = \sum \left( \frac{n_i}{N_i} \right)^{C^i} \tag{5}$$

where $n_i$ represents the number of the cycles of applied load at a given stress level $\sigma_i$, $N_i$ are the number of cycles to failure at $\sigma_i$, $C^i$ is the effect of the load sequence and $D$ s the total damage; after that, much research work has been performed [22–25] where the damage curve approach proposed by Manson and Halford [17,26] explains very well the effects of load sequences under two-level loading conditions, and their theory proceeds on the basis that the crack growth is the major evidence of damage [27]. The crack length growth is expressed by

$$a = a_0 + (0.18 - a_0) \left( \frac{n_a}{N_f} \right)^{\left( \frac{2}{3} \right) N_f^{0.4}} \tag{6}$$

where $n_a$ represents the applied cycles to reach a crack length of $a$, $N_f$ represents the number of cycles required to reach the fracture and $a_0$ is the characteristic defect length of

the material when $N_a/N_f = 0$. Thus, the cumulative damage $D$ is given by the cycle radio and the crack length.

$$D = \frac{1}{0.18}\left[a_0 + (0.18 - a_0)\left(\frac{n_a}{N_f}\right)^{\left(\frac{2}{3}\right)N_f^{0.4}}\right] \tag{7}$$

By applying Equation (7) in a connection of two levels sequence loading *A* and *B*, it is obtained,

$$D_A = \frac{1}{0.18}\left[a_0 + (0.18 - a_0)\left(\frac{n_1}{N_{f1}}\right)^{\left(\frac{2}{3}\right)N_{f1}^{0.4}}\right] \tag{8}$$

$$D_B = \frac{1}{0.18}\left[a_0 + (0.18 - a_0)\left(\frac{n_2}{N_{f2}}\right)^{\left(\frac{2}{3}\right)N_{f2}^{0.4}}\right] \tag{9}$$

Now, considering equal damage for the two load levels based on the theory of elasticity and the properties of the material [28], the equivalent damage cycle radio is,

$$\left(\frac{n_1}{N_{f1}}\right) = \left(\frac{n_2}{N_{f2}}\right)^{\left(\frac{N_{f2}}{N_{f1}}\right)^{0.4}} \tag{10}$$

Consequently, the damage curve is given by the power law equation,

$$D_i = \left(\frac{n_i}{N_{if}}\right)^{\left(\frac{N_{if}}{N_{if-1}}\right)^{0.4}} \tag{11}$$

where the exponent 0.4 represents a material constant characteristic cause–effect of deformation with cycles applied, and it determines the crack length growth from a microscopic perspective [23]. Now that the sequence is considered, let us present the proposed method where the random vibration PSD load effect is included in the nonlinear cumulative damage analysis.

Now, the random vibration PSD load effect is included in the nonlinear cumulative damage analysis.

### 3. Modified Nonlinear Fatigue Damage Accumulation Model Considering the Vibration Load Effect

As is described in [23], the damage curve model proposed by Manson and Halford presented prediction results close to experimental data. However, in particular for this paper, the load sequences and load interactions are induced by the effect of the random vibration. Then, to determine the fatigue damage accumulation, the damage curve approach model [17], as described by Equation (11), is modified to consider the random vibration PSD load effect, also.

Since in the damage curve model, the exponent value of 0.4 is based on the crack growth and it is constant, it does not consider the intensity effect of the loading change induced by a random vibration (PSD). Thus, to consider it, the exponent value 0.4 is replaced by $(\sigma i - 1_{vb}/\sigma i_{vb})$. This relation represents a nonlinear continuum damage function of the vibration bending stress induced.

That vibration bending stress is obtained from the PSD response acceleration function, as shown in Figure 1, where *A* is the speed for a given time *t* and *B* is the displacement for a given time *t*. $W_n$ is the natural frequency and $\Phi$ is the phase angle. Thus, the vector *C* represents the maximum amplitude (acceleration) of the movement, which, by its cyclical movement, induces fatigue damage.

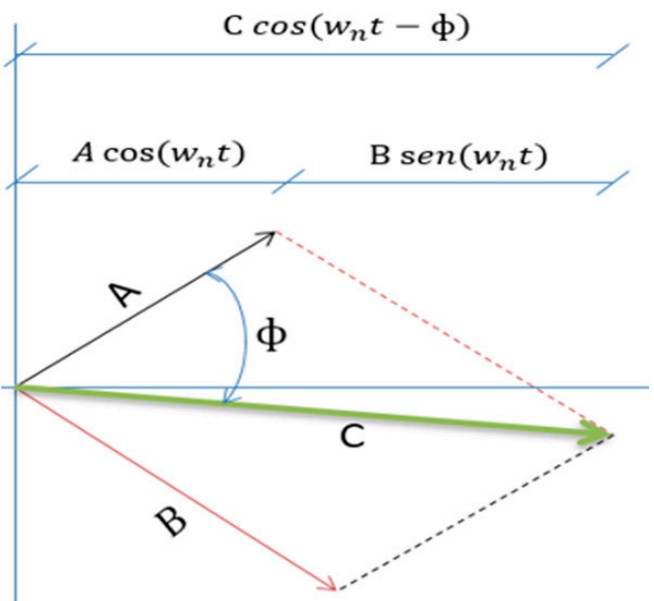

**Figure 1.** Vectorial representation of the movement.

The vector C and the phase angle Φ are obtained as

$$C = \sqrt{A^2 + B^2} \tag{12}$$

$$\Phi = \tan^{-1} \frac{B}{A} \tag{13}$$

Now, regarding the damage curve model, and considering the vibration bending stress, the proposed damage accumulation model under two level loading conditions is represented by,

$$D = \sum_{i=1}^{2} D_2 = \left[ \frac{n_2}{N_{2,f}} \right]^{\left( \frac{N_{2,f}}{N_{1,f}} \right)^{\left[ \frac{\sigma1_{vb}}{\sigma2_{vb}} \right]}} \tag{14}$$

This model presents the proposed nonlinear damage accumulation model that considers the effect of the sequence and interaction effect induced by a random vibration PSD. Here, it is highlighted that the total cycles to failure $N_i$ value is determined by using the Basquin's equation defined in Equation (25), and that it is determined by placing into Equation (25) the corresponding vibration bending stress $\sigma i_{vb}$. It is used instead of the constant tensile stress $\sigma i$ value that is obtained from the material's S-N diagram.

Thus, the proposed model for damage accumulation is applied, starting from calculating the bending stress caused by the effect of random vibration, as follows.

### 3.1. Calculating the Bending Stress

When a PSD vibration loading is applied to a component or product to cause the same damage as its dynamic environment will cause, a base input acceleration is applied. Consequently, a different response acceleration occurs, as represented in Figure 2 [29,30].

The reason for the difference between the input and response acceleration is due to the material's natural frequency and the effect of its own mass when it is exposed to the stress of vibration (PSD). Based on the generated displacement, it is measured as

$$Q = \frac{D_1}{D_2} \tag{15}$$

where $Q$ is an amplification factor, $D_1$ is the base (input) displacement and $D_2$ is the response displacement. Thus, the acceleration response is determined as

$$A_{res} = \frac{2\pi^2 F^2 D_2}{G} \tag{16}$$

where $F$ is the frequency applied by the PSD and $G$ is the gravity constant (9.81 $\frac{m}{s^2}$). Now, based on the generated moment (M), we determine the dynamic factor that includes the effect stress caused by the vibration PSD [29]. The reaction moment for bending stress, is given by,

$$M = F L \tag{17}$$

where $F$ corresponds to the effect of the mass of the component multiplied by the acceleration,

$$F = m A \tag{18}$$

Therefore, by considering the concept of effective mass ($m_e$), which represents the mass of an object that accelerates (vibrates) when an external force is exerted on it [29], its equation for the analysis is given by,

$$m_e \approx 0.225\rho L + m \tag{19}$$

where $\rho$ is the density of the component's material, $L$ is the length and m is the mass applied to the component. Thus, the bending moment $M_f$ is given by,

$$M_f = m_e \, A \, L \tag{20}$$

Consequently, the generated bending stress $\sigma_f$ is determined as

$$\sigma_f = \frac{K \, M_f \, C}{I} \tag{21}$$

Therefore, because Equation (21) includes Equation (20), then the scale factor of the stress induced by the stress generated by the vibration movement, here called dynamic load factor [2], is given as

$$\sigma_{dynamic} = \left( \frac{K m_e \hat{L} C}{I} \right) A \tag{22}$$

where $K$ is the stress concentration factor in the component, $C$ is the distance to the neutral axis, $\hat{L}$ is the distance from the fixed point of the component to the point of application of the mass, $A$ is the constant of gravity and $I$ is the moment of inertia given by,

$$I = \frac{1}{12} w t^3 \tag{23}$$

where $w$ is the width of the component and $t$ is the thickness of the material.

Once, from Equation (22), the dynamic factor $\sigma_{dynamic}$ is determined, and the response acceleration $A_{res}$ is determined from Equation (16), the vibration bending stress $\sigma_{vb}$ response to the vibration profile (base input acceleration) is determined as

$$\sigma_{vb} = \sigma_{dynamic} * A_{res} \tag{24}$$

Thus, finally, Equation (24) represents the bending stress that must be used in the vibration analysis to determine the expected useful life of the analyzed product that is being exposed to the environmental vibration.

Now, it is necessary to determine the cycles of vibration load applied $n_i$ at the given bending stress level $\sigma i_{vb}$.

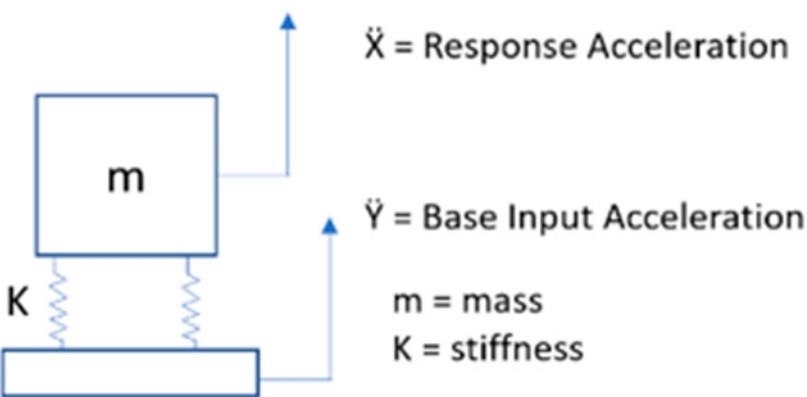

**Figure 2.** Response to a base acceleration.

### 3.2. Vibration Cycle Counting

To effectively determine the cycles $n_i$ of bending stress vibration, the Rainflow tool is used. This tool is validated by [31] Section 5.4.4. According to [10], this cycle counting method is as well represented as a variable amplitude cyclic loading as it is for random vibration. In Figure 3, it is shown how the rainflow cycle counting is performed. Doing this, (1) the sample time history of stress vs. time is obtained and (2) the time history diagram is rotated 90° clockwise and the counting of cycles at specific stress range is completed. The cycles' counting results are shown in Table 1.

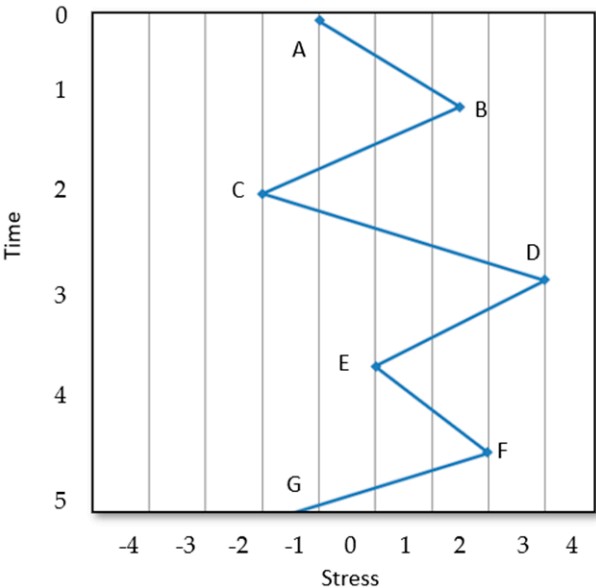

**Figure 3.** Rainflow diagram.

**Table 1.** Rainflow cycling counting.

| Rainflow Cycles | | |
|---|---|---|
| Path | Cycles | Stress Span |
| A–B | 0.5 | 2 |
| B–C | 0.5 | 3 |
| C–D | 0.5 | 5 |
| E–F | 1 | 2 |

Once the applied cycles $n_i$ are known by using the Rainflow method, they will be used to determine the fatigue damage caused by the random vibration's effect.

Now, it is required to estimate the total cycles $N_i$ at the given bending stress level $\sigma i_{vb}$ that the material's component can withstand.

### 3.3. Total Cycle Determination

Assuming that the S-N curve of the materials is stated by the Basquin's equation [5,32] as is in Equation (25),

$$N_i * \sigma_i^b = a^b \tag{25}$$

where $N_i$ is the total number of cycles that the element can sustain at a given stress level $\sigma_i$ and the constant material parameters a and b represent the intercept and the slope of the S-N curve, respectively. They are determined as

$$a = \frac{(fS_{ut})^2}{S_e} \tag{26}$$

$$b = -\frac{1}{3}log\left[\frac{fS_{ut}}{S_e}\right] \tag{27}$$

where $f$ is the fatigue resistance fraction, $S_{ut}$ is the ultimate resistance stress and $S_e$ is the endurance limit. Then, from Equation (25), the total cycles $N_i$ are determined as

$$N_i = \left(\frac{\sigma_i}{a}\right)^{\frac{1}{b}} \tag{28}$$

In mechanical design, the stress $\sigma_i$ given in Equation (28) is the equivalent stress or static stress. However, because here we are focused on the field of dynamic random vibration, then, in vibration analysis, the $\sigma_i$ is replaced by the vibration bending stress $\sigma i_{vb}$ determined in Section 3.1. Therefore, for vibration analysis, the total cycles are determined as

$$N_i = \left(\frac{\sigma i_{vb}}{a}\right)^{\frac{1}{b}} \tag{29}$$

Now that the cycles of vibration applied $n_i$ at the stress $\sigma i_{vb}$ and the total cycles $N_i$ that withstand at that stress $\sigma i_{vb}$ are known, we proceed to determine the accumulated fatigue damage that is caused by the random vibration loading, as follows.

## 4. Fatigue Damage Accumulation Procedure

The steps that illustrate how a PSD is used to calculate the fatigue damage of a mechanical component are presented in Figure 4.

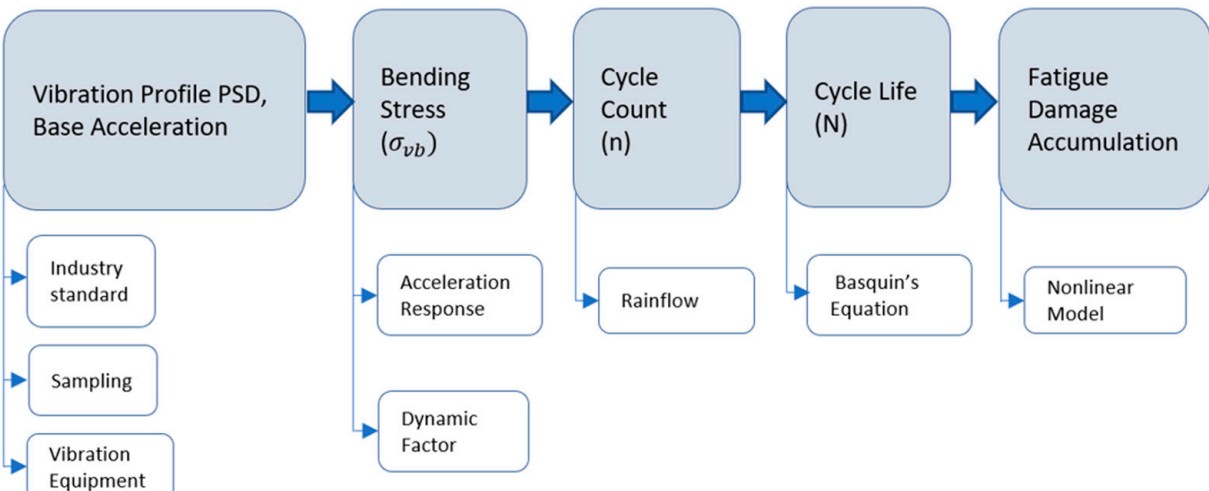

**Figure 4.** Process for estimating cumulative fatigue damage by random vibration loading.

The effectiveness of the above steps is illustrated in the experimental study case, as follows.

### 4.1. Study Case

We have a support that is subjected to a bending load (see Figure 5), and the support has the following features. It is cold drawn steel AISI 1025, with a modulus of elasticity E = 200 GPa, Poison's ratio $\gamma$ = 0.29, yield strength Sy = 430 MPa, ultimate tensile strength Sut = 510 MPa, endurance limit Se = 255 MPa, density $\rho$ = 7.9 g/cm$^3$, length L = 51 mm, Width W = 200 mm and a wall thickness of 3 mm. During its function, the component supports a load of 80 N, and its movement is free only in the vertical direction. Zone A (purple color) indicates that it is fixed, and zone B (red color) is the applied load. It is considered as a cantilevered beam and is submitted to an operating random vibration with an input PSD with the frequencies ranging from 10 to 55 Hz at an amplitude of 1.5 mm for a period of 2 h. The testing is carried out physically by using a vibration system.

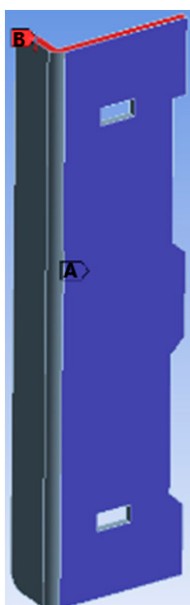

**Figure 5.** Support subjected to random vibration fatigue.

Thus, once the component has been submitted to the vibration fatigue by using the vibration system shaker with a vibration controller, it is required to calculate the bending stress.

### 4.2. Bending Stress

We proceed to determine the dynamic bending stress generated by the base input PSD. The base input PSD is presented in Table 2 and Figure 6a. In Figure 6b we show the corresponding time history synthesis of the PSD.

**Table 2.** Base input PSD.

| Frequency (Hz) | Accel. (G) | Accel. (G$^2$/Hz) |
|:---:|:---:|:---:|
| 10 | 0.210 | 0.004 |
| 20 | 0.666 | 0.022 |
| 30 | 1.383 | 0.063 |
| 40 | 2.359 | 0.139 |
| 50 | 3.900 | 0.300 |
| 55 | 4.664 | 0.400 |

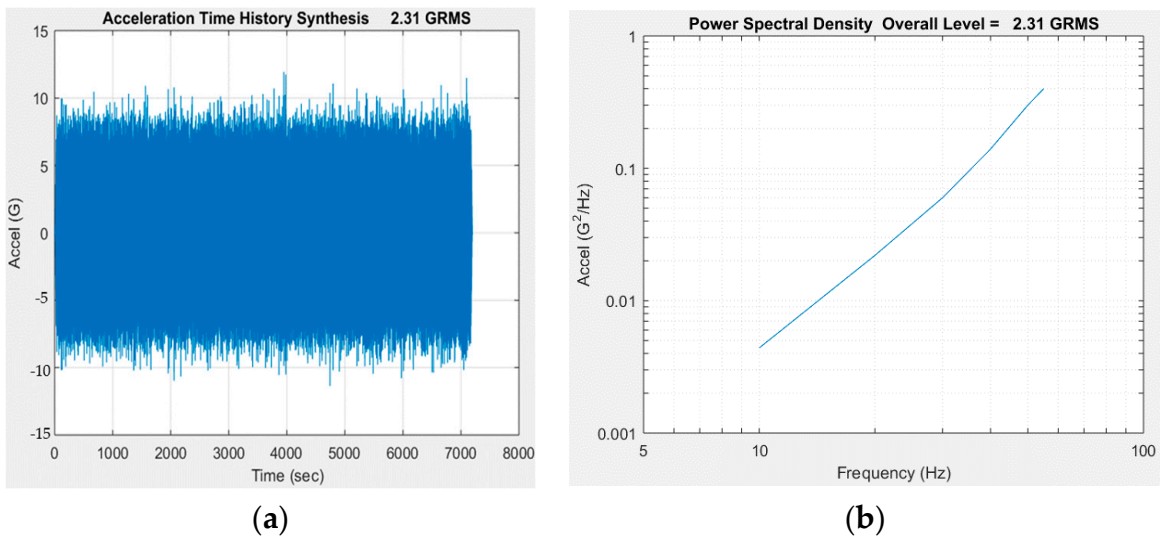

**Figure 6.** Base input PSD: (**a**) PSD 2.31 GRMS, (**b**) PSD time history synthesis.

In this case, the base input PSD is applied by a vibration shaker, and the acceleration response obtained is given in Table 3.

**Table 3.** Acceleration base input and acceleration response.

| Frequency (Hz) | Accel. Base Input (G) | Accel. Response (G) |
|---|---|---|
| 10 | 0.21 | 0.69 |
| 20 | 0.67 | 3.06 |
| 30 | 1.38 | 5.58 |
| 40 | 2.36 | 9.09 |
| 50 | 3.90 | 13.68 |
| 55 | 4.66 | 12.31 |

By comparing, in Figure 7, the input and response acceleration, we observed that the difference is due to the product's resonant frequency. From the analysis, we have the frequency of 50 Hz with an acceleration of 13.68 g, which is the one that more greatly affects the component.

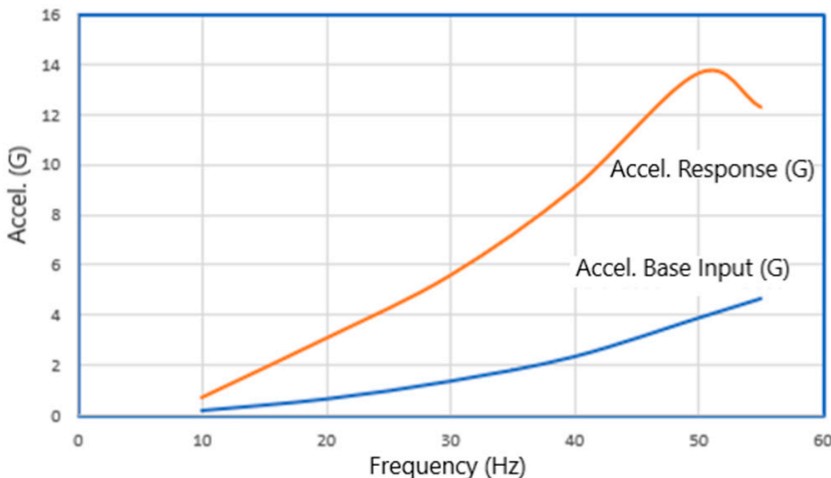

**Figure 7.** PSD of acceleration base input and acceleration response.

Now, we determine the dynamic factor that allows us to obtain the vibration stress that corresponds to the acceleration response obtained.

Since, from Equation (19) with $\rho = 0.285 \frac{lb}{in^3}$, $L = 2$ in and $m = 0.05 \frac{lb-s^2}{in}$, the effective mass is $m_e = 0.11 \frac{lb-s^2}{in}$, then, from Equation (22) with $K = 2.5$, $\hat{L} = 0.575$ in, $C = 0.06$ in, $I = 0.00114$ in$^4$ and $A = 9.81$ m/s$^2$, $m_e = 0.11 \frac{lb-s^2}{in}$ and the dynamic factor $\sigma_{dynamic.} = 3219$ Psi $= 3.22$ Ksi $= 22.22$ MPa.

Consequently, by using the acceleration response and the dynamic factor in Equation (24), the vibration bending stress value for each row of the PSD is obtained, as shown in Table 4.

**Table 4.** Bending stress results.

| Frequency (Hz) | Accel. Response (G) | Dynamic Factor $(\sigma_{dynamic})$ Equation (22) | Bending Stress $(\sigma_{vb})$ Equation (24) |
|---|---|---|---|
| 10 | 0.69 | | 15.33 |
| 20 | 3.06 | | 67.97 |
| 30 | 5.58 | 22.22 | 123.95 |
| 40 | 9.09 | | 201.91 |
| 50 | 13.68 | | 303.87 |
| 55 | 12.31 | | 273.44 |

Now, the cycles of vibration load applied $n_i$ at the given bending stress level $\sigma i_{vb}$ are determined.

### 4.3. Vibration Cycles Counting

In this case, we proceed to determine the stress Rainflow cycle count by using the software MATLAB with (ASTM E 1049-85), where the input data used are the frequency and acceleration response shown in Table 4. The results are shown in Table 5.

**Table 5.** Vibration cycling count.

| Frequency (Hz) | Vibration Cycling ($n_i$) |
|---|---|
| 10 | 70,292 |
| 20 | 140,210 |
| 30 | 92,650 |
| 40 | 10,869 |
| 50 | 3010 |
| 55 | 816 |

Next, the total cycles $N_i$ at the given bending stress level $\sigma i_{vb}$ are determined.

### 4.4. Total Cycles Determination

To determine the total cycles $N_i$ by using Equation (29), it is required to calculate the constants $a$ and $b$ by using the material's properties in Equations (26) and (27), as follows,

$$a = \frac{(0.9x510 \; MPa)^2}{255 \; MPa} = 826.2$$

$$b = -\frac{1}{3}log\left[\frac{0.9x380 \; MPa}{255 \; MPa}\right] = -0.085$$

Therefore, the results of total cycles for the vibration PSD applied are shown in Table 6.

In the next section, the fatigue damage accumulation caused by the random vibration is determined by using the proposed nonlinear model.

**Table 6.** Total cycles.

| Frequency (Hz) | Bending Stress ($\sigma_{vb}$) Equation (24) | Total Cycle ($N_i$) Equation (29) |
|---|---|---|
| 10 | 15.33 | $2.24 \times 10^{20}$ |
| 20 | 67.97 | $5.60 \times 10^{12}$ |
| 30 | 123.95 | $4.81 \times 10^{9}$ |
| 40 | 201.91 | $1.55 \times 10^{7}$ |
| 50 | 303.87 | $1.27 \times 10^{5}$ |
| 55 | 273.44 | $4.40 \times 10^{5}$ |

*4.5. Fatigue Damage Accumulation*

Here, to determine the vibration fatigue damage caused by the PSD loading applied to the mechanical component, Equation (14) is applied. That equation determines the fatigue damage accumulation as a nonlinear cumulative process by using the bending vibration stress ($\sigma i_{vb}$), the vibration cycles ($n_i$), and the vibration total cycles ($N_i$). The results are shown in Table 7.

**Table 7.** Cumulative damage calculation using proposed nonlinear model.

| | 10 Hz | 20 Hz | 30 Hz | 40 Hz | 50 Hz | 55 Hz |
|---|---|---|---|---|---|---|
| **Block No.** | $D_1$ | $D_{1+2}$ | $D_{1+2+3}$ | $D_{1+2+3+4}$ | $D_{1+2+3+4+5}$ | $D_{1+2+3+4+5+6}$ |
| 1 (2 h) | $3.14 \times 10^{-16}$ | $3.14 \times 10^{-16}$ | $3.16 \times 10^{-16}$ | $3.44 \times 10^{-16}$ | $2.36 \times 10^{-2}$ | $2.41 \times 10^{-2}$ |
| 2 (2 h) | $2.41 \times 10^{-2}$ | $2.41 \times 10^{-2}$ | $2.42 \times 10^{-2}$ | $2.46 \times 10^{-2}$ | $4.83 \times 10^{-2}$ | $4.90 \times 10^{-2}$ |
| 3 (2 h) | $4.90 \times 10^{-2}$ | $4.90 \times 10^{-2}$ | $4.93 \times 10^{-2}$ | $5.01 \times 10^{-2}$ | $7.37 \times 10^{-2}$ | $7.48 \times 10^{-2}$ |
| 4 (2 h) | $7.48 \times 10^{-2}$ | $7.48 \times 10^{-2}$ | $7.53 \times 10^{-2}$ | $7.64 \times 10^{-2}$ | $1.00 \times 10^{-1}$ | $1.01 \times 10^{-1}$ |
| 5 (2 h) | $1.01 \times 10^{-1}$ | $1.01 \times 10^{-1}$ | $1.02 \times 10^{-1}$ | $1.04 \times 10^{-1}$ | $1.27 \times 10^{-1}$ | $1.29 \times 10^{-1}$ |
| 6 (2 h) | $1.29 \times 10^{-1}$ | $1.29 \times 10^{-1}$ | $1.30 \times 10^{-1}$ | $1.32 \times 10^{-1}$ | $1.55 \times 10^{-1}$ | $1.57 \times 10^{-1}$ |
| 7 (2 h) | $1.57 \times 10^{-1}$ | $1.57 \times 10^{-1}$ | $1.58 \times 10^{-1}$ | $1.60 \times 10^{-1}$ | $1.84 \times 10^{-1}$ | $1.86 \times 10^{-1}$ |
| 8 (2 h) | $1.86 \times 10^{-1}$ | $1.86 \times 10^{-1}$ | $1.87 \times 10^{-1}$ | $1.90 \times 10^{-1}$ | $2.14 \times 10^{-1}$ | $2.16 \times 10^{-1}$ |
| 9 (2 h) | $2.16 \times 10^{-1}$ | $2.16 \times 10^{-1}$ | $2.17 \times 10^{-1}$ | $2.20 \times 10^{-1}$ | $2.44 \times 10^{-1}$ | $2.47 \times 10^{-1}$ |
| 10 (2 h) | $2.47 \times 10^{-1}$ | $2.47 \times 10^{-1}$ | $2.48 \times 10^{-1}$ | $2.52 \times 10^{-1}$ | $2.75 \times 10^{-1}$ | $2.78 \times 10^{-1}$ |
| 11 (2 h) | $2.78 \times 10^{-1}$ | $2.78 \times 10^{-1}$ | $2.80 \times 10^{-1}$ | $2.84 \times 10^{-1}$ | $3.08 \times 10^{-1}$ | $3.11 \times 10^{-1}$ |
| 12 (2 h) | $3.11 \times 10^{-1}$ | $3.11 \times 10^{-1}$ | $3.13 \times 10^{-1}$ | $3.17 \times 10^{-1}$ | $3.41 \times 10^{-1}$ | $3.44 \times 10^{-1}$ |
| 13 (2 h) | $3.44 \times 10^{-1}$ | $3.44 \times 10^{-1}$ | $3.46 \times 10^{-1}$ | $3.51 \times 10^{-1}$ | $3.75 \times 10^{-1}$ | $3.78 \times 10^{-1}$ |
| 14 (2 h) | $3.78 \times 10^{-1}$ | $3.78 \times 10^{-1}$ | $3.81 \times 10^{-1}$ | $3.86 \times 10^{-1}$ | $4.10 \times 10^{-1}$ | $4.13 \times 10^{-1}$ |
| 15 (2 h) | $4.13 \times 10^{-1}$ | $4.13 \times 10^{-1}$ | $4.16 \times 10^{-1}$ | $4.22 \times 10^{-1}$ | $4.45 \times 10^{-1}$ | $4.49 \times 10^{-1}$ |
| 16 (2 h) | $4.49 \times 10^{-1}$ | $4.49 \times 10^{-1}$ | $4.52 \times 10^{-1}$ | $4.59 \times 10^{-1}$ | $4.82 \times 10^{-1}$ | $4.87 \times 10^{-1}$ |
| 17 (2 h) | $4.87 \times 10^{-1}$ | $4.87 \times 10^{-1}$ | $4.90 \times 10^{-1}$ | $4.96 \times 10^{-1}$ | $5.20 \times 10^{-1}$ | $5.25 \times 10^{-1}$ |
| 18 (2 h) | $5.25 \times 10^{-1}$ | $5.25 \times 10^{-1}$ | $5.28 \times 10^{-1}$ | $5.35 \times 10^{-1}$ | $5.59 \times 10^{-1}$ | $5.25 \times 10^{-1}$ |
| 19 (2 h) | $5.25 \times 10^{-1}$ | $5.25 \times 10^{-1}$ | $5.67 \times 10^{-1}$ | $5.75 \times 10^{-1}$ | $5.99 \times 10^{-1}$ | $6.04 \times 10^{-1}$ |
| 20 (2 h) | $6.04 \times 10^{-1}$ | $6.04 \times 10^{-1}$ | $6.07 \times 10^{-1}$ | $6.16 \times 10^{-1}$ | $6.39 \times 10^{-1}$ | $6.45 \times 10^{-1}$ |
| 21 (2 h) | $6.45 \times 10^{-1}$ | $6.45 \times 10^{-1}$ | $6.49 \times 10^{-1}$ | $6.58 \times 10^{-1}$ | $6.81 \times 10^{-1}$ | $6.87 \times 10^{-1}$ |
| **22 (2 h)** | $6.87 \times 10^{-1}$ | $6.87 \times 10^{-1}$ | $6.91 \times 10^{-1}$ | $\mathbf{7.01 \times 10^{-1}}$ | $7.24 \times 10^{-1}$ | $7.30 \times 10^{-1}$ |
| 23 (2 h) | $7.30 \times 10^{-1}$ | $7.30 \times 10^{-1}$ | $7.35 \times 10^{-1}$ | $7.45 \times 10^{-1}$ | $7.68 \times 10^{-1}$ | $7.74 \times 10^{-1}$ |
| 24 (2 h) | $7.74 \times 10^{-1}$ | $7.74 \times 10^{-1}$ | $7.79 \times 10^{-1}$ | $7.90 \times 10^{-1}$ | $8.13 \times 10^{-1}$ | $8.20 \times 10^{-1}$ |
| 25 (2 h) | $8.20 \times 10^{-1}$ | $8.20 \times 10^{-1}$ | $8.25 \times 10^{-1}$ | $8.36 \times 10^{-1}$ | $8.60 \times 10^{-1}$ | $8.66 \times 10^{-1}$ |
| 26 (2 h) | $8.66 \times 10^{-1}$ | $8.66 \times 10^{-1}$ | $8.72 \times 10^{-1}$ | $8.83 \times 10^{-1}$ | $9.07 \times 10^{-1}$ | $9.14 \times 10^{-1}$ |
| 27 (2 h) | $9.14 \times 10^{-1}$ | $9.14 \times 10^{-1}$ | $9.20 \times 10^{-1}$ | $9.32 \times 10^{-1}$ | $9.56 \times 10^{-1}$ | $9.63 \times 10^{-1}$ |
| **28 (2 h)** | $9.63 \times 10^{-1}$ | $9.63 \times 10^{-1}$ | $9.69 \times 10^{-1}$ | $9.82 \times 10^{-1}$ | $\mathbf{1.01 \times 10^{0}}$ | $1.01 \times 10^{0}$ |

From the results given in Table 7, we noticed that the damage $D = 0.70$ is reached at block 22. Some industries act when their products reach this quantity of damage, and the damage $D = 1.0$ (fatigue failure) is reached at block 28 see Figure 8.

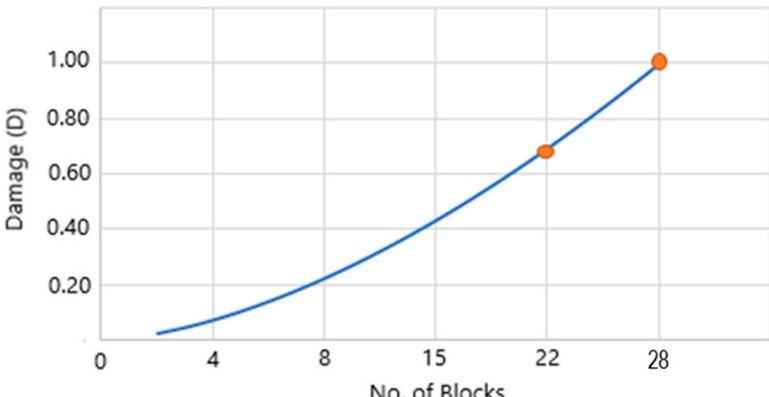

**Figure 8.** Damage curve.

The curve in Figure 8 represents the damage line relationship of the support submitted to random vibration loading.

From Table 7, it is seen that the frequency that most affects the mechanical component and induces damage is the frequency of 50 Hz. The reason is explained as follows:

The natural frequency $W_n$ of the component is determined by

$$W_n = \sqrt{\frac{3EI}{ml^3}} \tag{30}$$

where $E$ is the modules of elasticity in $\frac{lb}{in^2}$, $I$ is the moment of inertia in $in^4$ given by Equation (23), $m$ is the load mass in $\frac{lb-s^2}{in}$ determined by Equation (19) and $l$ is the component's length in inches. With these data, $W_n$ is

$$W_n = \sqrt{\frac{3\left(29007548\,\frac{lb}{in^2}\right)\left(0.00114\,in^4\right)}{\left(0.11\,\frac{lb-s^2}{in}\right)(2\,in)^3}} = 311.87\,\frac{rad}{s}$$

Consequently, from Equation (4), the natural frequency $f_n$ is

$$f_n = \frac{311.87}{2\pi} \approx 50\,Hz$$

Thus, since the material's natural frequency is $f_n = 50$ Hz, then, because the frequencies of the applied PSD range from 10 to 55 Hz, at the moment, they coincide with the resonant frequency presented. This can be seen in Figure 8 where, due to the resonant frequency, the maximum acceleration that induces damage is reached at 50 Hz.

As a general conclusion, in the proposed nonlinear random vibration method, the damage accumulation model can be applied by using the vibration response acceleration PSD to obtain the bending stress induced and, therefore, determine the fatigue damage accumulation. However, it is highlighted that the efficiency of the proposed method depends on the accuracy at which the response acceleration PSD is obtained. In this case, we used vibration shaker equipment and its accelerometer, but for the design and prototype phase, it can be performed by using a software simulation. In general, the proposed method can be applied to any mechanical component analysis where the response acceleration $A_{res}$ is known.

## 5. Conclusions

(1) Since the inputs for the nonlinear model are the response acceleration's $A_{res}$ values, and because, by using the dynamic factor $\sigma_{dynamic}$ , the vibration bending stresses' $\sigma_{vb}$ values can be determined, then the proposed model can be applied in any me-

chanical component analysis submitted to random vibration forces where $A_{res}$ values are known.

(2) Since the dynamic factor $\sigma_{dynamic}$ allows us to obtain stress units from acceleration units, then the vibration bending stress $\sigma_{vb}$ values can be used in the proposed model Equation (14) to calculate the fatigue damage accumulation induced by the random vibration.

(3) The constant exponent parameter value of 0.4 in the Manson–Halford model has been replaced by a vibration bending stress relation that considers the effect sequence and interaction loads induced by the applied PSD.

(4) As a result of the mechanical component case study, the model proposed to predict the fatigue damage accumulation shows good agreement with the reported experimental data. The model predicts 28 cycles of load to reach the failure ($D = 1$), and the experiment data collected from the vibration shaker showing, after 28 cycles of load, the deformation on the component's material can be considered as a failure.

(5) The efficiency of the model proposed offers the advantage that even though random vibration (PSD) provides a complex loading history, based on the corresponding PSD, it is possible to obtain the fatigue damage accumulation through general analysis.

(6) Since the proposed method is based on the bending stress induced by random vibration PSD function and its response acceleration, then, knowing that the fatigue damage accumulation can also be analyzed from a strain and crack growth propagation point of view, it seems that the proposed model could also be used in those cases, but more research must be undertaken.

**Author Contributions:** Conceptualization, J.M.B.-C., M.R.P.-M. and A.M.; methodology, J.M.B.-C. and M.R.P.-M.; data analysis, J.M.B.-C. and A.M.; writing—original draft preparation, J.M.B.-C. and M.R.P.-M.; writing—review and editing, J.M.B.-C., M.R.P.-M. and R.C.T.-V.; supervision, M.R.P.-M.; funding acquisition, J.M.B.-C. and R.C.T.-V. All authors have read and agreed to the published version of the manuscript.

**Funding:** The research received no external funding.

**Institutional Review Board Statement:** Not applicable.

**Informed Consent Statement:** Not applicable.

**Data Availability Statement:** Not applicable.

**Conflicts of Interest:** The authors declare no conflict of interest.

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
