# Peer review of "Random Vibration Fatigue Analysis Using a Nonlinear Cumulative Damage Model"

_applsci, doi:10.3390/app12094310_

Round 1

Reviewer 1 Report

The subject is very intersting but materials presented in paper is not good 

organized.
At first it sgoulb prezented linear models, next nonlinaer cumaltive models and next
new model. The Authors should read and cited such paper conecting with nonlinaer models:

Calderon-Uriszar-Aldaca, I. and Biezma, M. V., “A Plain Linear Rule for Fatigue Analysis under Natural Loading

Considering the Sequence Effect,” Int. J. Fatigue, Vol. 103, 2017, pp. 386–394

Pompetzki, M., Topper, T., DuQuesnay, D., and Yu, M., “Effect of Compressive Underloads and Tensile Overloads on Fatigue Damage Accumulation in 2024-T351 Aluminum,” J. Test. Eval., Vol. 18, No. 1, 1990, pp. 53–61,

Pawliczek, R. and Prażmowski, M., “Study on Material Property Changes of Mild Steel S355 Caused by Block Loads with Varying Mean Stress,” Int. J. Fatigue, Vol. 80, 2015, pp. 171–177

Szala, G. and Ligaj, B., “Application of Hybrid Method in Calculation of Fatigue Life for C45 Steel (1045 Steel) Structural Components,” Int. J. Fatigue, Vol. 91, Part 1, 2016, pp. 39–49

Garcia, S., Amrouche, A., Mesmacque, G., Decoopman, X., and Rubio, C., “Fatigue Damage Accumulation of Cold Expanded Hole in Aluminum Alloys Subjected to Block Loading,” Int. J. Fatigue, Vol. 27, Nos. 10–12, 2005,pp. 1347–1353

Troshchenko, V. T., Dragan, V. I., and Semenyuk, S. M., “Fatigue Damage Accumulation in Aluminium and Titanium Alloys Subjected to Block Program Loading under Conditions of Stress Concentration and Fretting,” Int. J. Fatigue, Vol. 21, No. 3, 1999, pp. 271–279,

Taheri, S., Vincent, L., and Le-roux, J.-C., “A New Model for Fatigue Damage Accumulation of Austenitic Stainless Steel under Variable Amplitude Loading,” Procedia Eng., Vol. 66, 2013, pp. 575–586,

Yuan, R., Li, H., Huang, H. Z., Zhu, S. P., and Li, Y. F., “A New Non-Linear Continuum Damage Mechanics Model for the Fatigue Life Prediction under Variable Loading,” Mechanics, Vol. 19, No. 5, 2013, pp. 506–511,

Kwofie, S. and Rahbar, N., “A Fatigue Driving Stress Approach to Damage and Life Prediction under Variable Amplitude Loading,” Int. J. Damage Mech., Vol. 22, No. 3, 2013, pp. 393–404,

Epaarachchi, J. A. and Clausen, P. D., “An Empirical Model for Fatigue Behavior Prediction of Glass Fibre-Reinforced

Plastic Composites for Various Stress Ratios and Test Frequencies,” Composites Part A, Vol. 34, No. 4, 2003, pp. 313–326,

Anes, V., Caxias, J., Freitas, M., and Reis, L., “Fatigue Damage Assessment under Random and Variable Amplitude Multiaxial Loading Conditions in Structural Steels,” Int. J. Fatigue, Vol. 100, Part 2, 2017, pp. 591–601,

Benkabouche, S., Guechichi, H., Amrouche, A., and Benkhettab, M., “A Modified Nonlinear Fatigue Damage Accumulation Model under Multiaxial Variable Amplitude Loading,” Int. J. Mech. Sci., Vol. 100, 2015, pp. 180–194,

Or a new model:

Bohm E., Kurek M., Łagoda T., Łagoda K., The use a power law function for fatigue life estimation for block loads, Solid State Phenomena, vol.250, 2016, pp.1-9

Böhm, M. Kurek, and T. Łagoda, Fatigue Damage Accumulation Model of 6082-T6 Aluminum Alloy in Conditions of Block Bending and Torsion, Journal of Testing and Evaluation 48, no. 6 (November/December 2020), pp.4416–4434

  • in absreact it not should be cited Eqs
  • - p2 l71 - it is cited Eq 24 but this Eq is on the next pages
  • - p.4 ls134-140. Plaese add more informations on exponent 0.4 -  which determines?
    - p.6, l.214 (also p.11) - please cut information about "in". It should be used SI standard
    - p.8 - Basquin Eq is defined in a strange way
    - p.9, l.309 - What does mean "L"?
    - in bendind - the stress is nominal or elasto-plastic?
    - l.374 it is written "G". It shpuld be "g"
    - l.464 - it is rad/seg. It should be rad/sec?
    - ref.[11] - It should be Osgood (not OSGOOD)

Reviewer 2 Report

The authors performed random vibration fatigue analysis based on a nonlinear cumulative damage model. The exponent value of 0.4 from the Manson-Halford curve damage model was replaced by a vibration bending stress relation. In general, I think that the idea is new and the paper is well written.

I have the following comments or suggestions which might help to improve the quality of this paper.

  1. In Introduction, please discuss more on why this paper is way better than existing researches in performing the fatigue analysis.
  2. Could the authors discuss more on why the well-known DLDR method is used, i.e. why DLDR is more appropriate here than other nonlinear methods?
  3. Could the authors discuss more on the results shown in Figure 8, or go further and develop an empirical model?
  4. Some typos and grammar errors still exist. Please check the whole paper carefully.

Round 2

Reviewer 1 Report

It may be published as is